# Inclusion of environmentally themed search terms improves Elastic net regression nowcasts of regional Lyme disease rates

Eric Kontowicz[1,2], Grant Brown[3], James Torner[1], Margaret Carrel[4], Kelly K. Baker[5], Christine A. Petersen [1,2,6] *

1 Department of Epidemiology, College of Public Health, University of Iowa, Iowa City, Iowa, United States of America, 2 Center for Emerging Infectious Diseases, University of Iowa Research Park, Coralville, Iowa, United States of America, 3 Department of Biostatistics, College of Public Health, University of Iowa, Iowa City, Iowa, United States of America, 4 Department of Geographical and Sustainability Sciences, College of Liberal Arts & Sciences, University of Iowa, Iowa City, Iowa, United States of America, 5 Department of Occupational and Environmental Health, College of Public Health, University of Iowa, Iowa City, United States of America, 6 Immunology Program, Carver College of Medicine, University of Iowa, Iowa City, Iowa, United States of America

* christine-petersen@uiowa.edu

**Data Availability Statement:** We have uploaded our relevant code and data files into github. https://

## Abstract

Lyme disease is the most widely reported vector-borne disease in the United States. 95% of confirmed human cases are reported in the Northeast and upper Midwest (25,778 total confirmed cases from Northeast and upper Midwest / 27,203 total US confirmed cases). Human cases typically occur in the spring and summer months when an infected nymph *Ixodid* tick takes a blood meal. Current federal surveillance strategies report data on an annual basis, leading to nearly a year lag in national data reporting. These lags in reporting make it difficult for public health agencies to assess and plan for the current burden of Lyme disease. Implementation of a nowcasting model, using historical data to predict current trends, provides a means for public health agencies to evaluate current Lyme disease burden and make timely priority-based budgeting decisions. The objective of the study was to develop and compare the performance of nowcasting models using free data from Google Trends and Centers of Disease Control and Prevention surveillance reports. We developed two sets of elastic net models for five regions of the United States: 1. Using only monthly proportional hit data from the 21 disease symptoms and tick related terms, and 2. Using monthly proportional hit data from terms identified via Google correlate and the disease symptom and vector terms. Elastic net models using the full-term list were highly accurate (Root Mean Square Error: 0.74, Mean Absolute Error: 0.52, $R^2$: 0.97) for four of the five regions of the United States and improved accuracy 1.33-fold while reducing error 0.5-fold compared to predictions from models using disease symptom and vector terms alone. Many of the terms included and found to be important for model performance were environmentally related. These models can be implemented to help local and state public health agencies accurately monitor Lyme disease burden during times of reporting lag from federal public health reporting agencies.

github.com/ekontowicz/Lyme-disease-Elastic-Net-regression-Nowcasting.

**Funding:** The authors received no specific funding for this work.

**Competing interests:** The authors have declared that no competing interest exists.

## Introduction

Lyme disease is the most widely reported vector-borne disease in the United States [1], with 95% (25,778 total confirmed cases from Northeast and upper Midwest / 27,203 total US confirmed cases) of human cases occurring in the Northeast and upper Midwest [2]. *Borrelia burgdorferi sensu lato* (including *Borrelia mayonii*, hereafter *B. burgdorferi*) is the causative agent of Lyme disease. It is transmitted to people predominantly when nymph or, to a lesser extent, adult ticks infected with *B. burgdorferi* take a blood meal [3, 4]. Hard to detect nymphal *Ixodes* ticks quest for blood meals during spring and early summer months. People are at greatest risk of contracting Lyme disease during and immediately following this time [5–9] when spending time in the environment for either work or recreation [2]. Areas with sandy soil and wooded vegetation are environmental factors associated with higher tick densities [10]. With increased geographic spread of Lyme disease, there has been increased incidence since 2000 [11]. Lyme disease has a large economic burden on patients and their surrounding communities [12, 13].

Surveillance of Lyme disease in the United States requires participation from many different areas of the health care system [14]. This surveillance relies on case reports from physicians, lab reports from diagnostic labs and collation of this data as cases by local and state health departments. These case reports are forwarded to the Centers of Disease Control and Prevention (CDC), which then aggregates the data and produces summary reports on national Lyme disease incidence. Due to differences in reporting from states and localities, compilation of data at the federal level can take several years, resulting in a time lag for release of nationwide surveillance and summary reports. This lag in federal reporting has been problematic for local health departments (LHDs), as they must predict current and emerging public health needs based on Federal data that is several years old [15]. LHDs not only play a vital role in surveillance of Lyme disease, but also help mitigate disease incidence through the implementation of local interventions. Funded prevention efforts/campaigns by LHDs can have a positive effect on health in communities [16]. Unfortunately, there are often many important competing health priorities in communities. As such, LHDs must make critical decisions to allocate their limited fund to areas of highest need. Modeling methods that accurately nowcast, or predict the present, Lyme disease incidence in a region would allow for better planning on the part of LHDs to allocate their efforts. Using statistical learning methods for nowcasting can also discover, or highlight, patterns that are associated with disease and can be used to generate future hypotheses.

Infodemiology is an emerging area of science research focused on utilizing information from an electronic medium (typically the internet) with an aim to inform or improve public health [17, 18]. Examples of infodemiology include monitoring Twitter, facebook posts, or Instagram hashtags for syndromic surveillance, identifying access or misinformation about vaccination or other public health initiatives, and measuring the effectiveness of public health education messages [17]. In developed countries approximately 94%of younger generations have access to and use the internet according to the International Telecommunication Union [19]. This increase in internet usage has changed the way individuals seek and receive health information and provides researchers with new opportunities to improve disease prediction and public information [20].

Usage of non-traditional indicators of disease spread, like Google search traffic history, has gained credibility from public health audiences [21, 22]. Google search data has been used with a variety of mathematical and statistical models to predict obesity rates, unemployment rates, and infectious diseases with varying levels of accuracy [23–26]. The principal insight of these approaches is that search data is available at a wide temporal and geographical scale, and such queries may be correlated with a phenomena or disease process of interest or human

behaviors [27]. This correlation can be leveraged to make predictions of current or future health outcome rates. In addition, relative frequencies of search terms may generate interesting hypotheses concerning human behaviors and their relationship with disease outcomes.

Given the complex and potentially high dimensional nature of search data, statistical and machine learning tools are a natural fit for model development. There are a variety of parametric and non-parametric statistical learning approaches used in the literature for infectious disease prediction as discussed in a recent review [28]. In this work, we do not provide a comprehensive review of such options, but rather seek to demonstrate that nowcasting is a promising opportunity for Lyme disease specifically. For this reason, we employ Elastic net regression. Elastic net regression provides a flexible parametric approach which strikes a compromise between the L1 and L2 penalties of Least Absolute Shrinkage and Selection Operator (LASSO) and ridge regression, respectively. It is also computationally straightforward, being easily employed on modest hardware. An additional advantage of elastic net regression is the grouping effect, where strongly correlated features tend to remain or be excluded from the model together [29].

In this study, we built elastic net regression models capable of nowcasting Lyme disease rates in five different regions of the United States. We developed two models for each region, 1. Using search traffic data from only disease name, symptom and vector related terms and 2. Using search traffic from terms identified via Google Correlate™ in addition to disease name, symptom and vector related terms to identify trends using information recently sought by the general public on the disease, it's symptoms, and correlated terms [30, 31]. We hypothesized that nowcasting models would have better predictive accuracy and lower error when using a full list of search terms that the average person would search compared to models that only use terms related to disease name, symptom and vectors of Lyme disease. Further, the three most important terms from accurate models will be potential exposure/location themed and their search patterns will align temporally with the timing of Lyme disease incidence in endemic areas, the Northeast and Midwest, and less well in non-endemic areas, the Southwest and West.

## Materials and methods

### Outcome data

All Lyme disease incidence data for this study was provided by the United States Centers for Disease Control and Prevention (CDC) (https://www.cdc.gov/lyme/stats/tables.html). In 2008, the CDC switched to a Suspected, Probable, or Confirmed case reporting approach. Cases were considered confirmed if an individual presents with erythema migrans (bullseye rash) and with a known exposure, a case of erythema migrans with laboratory evidence of infection and without known exposure, or a case with at least one late manifestation that has laboratory evidence of infection. Any other case of physician-diagnosed Lyme disease that has laboratory evidence of infection were considered probable cases. Both confirmed and probable case definitions were included to provide a more sensitive and inclusive criterion. Laboratory evidence of infection in both definitions allowed for strong confidence in a Lyme disease diagnosis. Even so, heterogeneity remains in reporting strategy; between 2015 and 2016, Massachusetts changed their reporting strategy to only report laboratory confirmed cases to the CDC. Only reporting laboratory confirmed cases is likely to lead to underreporting of the true burden of disease [32]. Lyme disease incidence is reported by the CDC on a per county of diagnosis for each US state. For the purposes of this study, we aggregated these counts to state and month based on date of diagnosis. Next, regional incidence rates were calculated for five different regions: Northeast, Midwest, Southeast, Southwest and West (**Fig 1**). Regions were developed

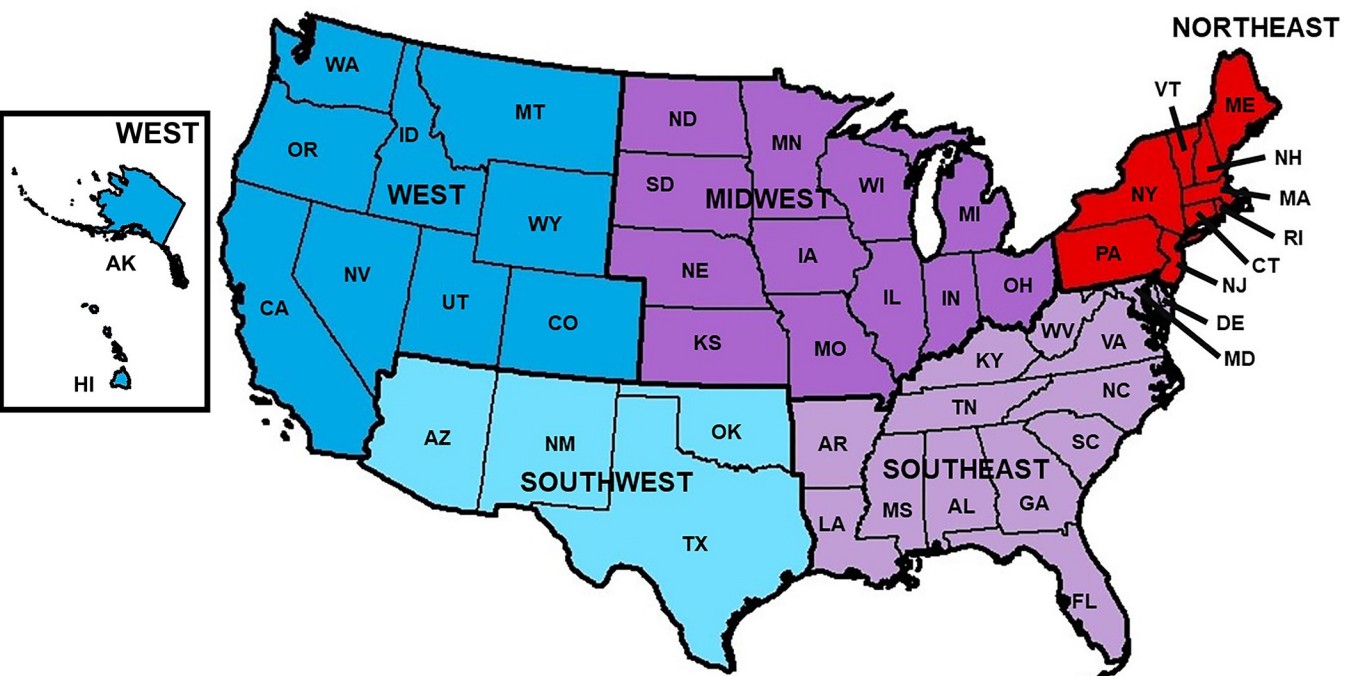

**Fig 1. Regions of the United States.** United States divided into 5 different regions by the geography division of the U.S. census bureau (Northeast, Midwest, Southeast, Southwest, and West) used to calculate regional Lyme incidence, and regional search term data. Map created using ArcGIS software using US census bureau.

as a hybrid of known high incidence regions and the US Census regions [14, 33]. Regional monthly Lyme disease incidence rates were calculated using combined state level population data from the 2010 US Census. Data was split into training and hold-out sets; models were fit on observations between February 2004 and December 2014 and validated on the hold-out observations which had available surveillance data from January 2015 to December 2017.

## Google search term data

Regional Lyme disease incidence trends from the training period were used with Google Correlate[TM] to identify the top 100 correlated search terms on which monthly proportional search hit data was later collected via Google Trends[TM] [34]. Google Correlate[TM] was not able to identify terms at state levels. These correlations can only be made on a nationwide basis for a submitted time series. Thus, we were not able to limit our search term identification by region. However, using regional Lyme disease time series data, provided many regionally specific terms in the top 100 correlated terms for each region (S1 Table). Strong correlation was determined when the correlation value was greater than 0.8, moderate if correlation value was between 0.5 and 0.8, and poor when less than 0.5. Strength of correlation was determined by correlation value (r) and significance of correlation was determined by p-value $< 0.05$.

Google Correlate[TM] implements an Approximate Nearest Neighbor (ANN) system to identify candidate search terms that matched similar temporal trends from supplied data. This system implemented a two-pass hash-base system. The first pass computed the approximate distance from the supplied time series to a hash of each series in Google's database [34]. The second pass computed the exact distance function using the top results supplied from the first pass [34]. For each region, the 100 terms identified from Google Correlate[TM] and the 21 Lyme disease symptom and *Ixodid-* vector related terms were entered into *gtrendsR* (an interface to

obtain Google Trends[TM] queries via R) [35] to collect proportional monthly search hit data for each term per region [35, 36]. Data was collected for regional search traffic in a systematic way similar to Mavragani and Ochoa (2019) [37]. Keywords were identified via Google Correlate[TM] plus 21 Lyme disease symptoms and *Ixodid*- vector related terms. These terms were entered into Google Trends[TM] via *gtrendsR* without alteration or use of quotations or combination. Regional search hit data was collected at the overall state level (including metropolitan, urban, suburban and rural searches) for each term and averaged to regional aggregates. Geographic regions were: Northeast, Southeast, Midwest, Southwest and West (**Fig 1**). The selected period of search data was 2004 to 2019, collected at monthly to match temporal aggregation of Lyme disease data. Search categories were not implemented in this research. Search hit data was collected between September 18, 2019 and September 26, 2019. This was then used as feature data for nowcasting Lyme disease incidence trends [38, 39].

## Modeling

For each region, two groups of elastic net regression models were fit for comparison: 1. a model using only monthly proportional hit data from the 21 disease symptoms and tick related terms list, and 2. a model using monthly proportional hit data from terms identified via Google Correlate[TM] in combination with the disease symptom and tick term list (this will be referred to as the full-term list for the remainder of the paper). The training data was from February 2004 through December 2014. To help prevent overfitting we implemented a rolling training window for the statistical learning process with a twelve-month learning window and one month validation window. To further address the potential for overfitting, we excluded data between January 2015 until December 2017 from the model training process. The hold-out data set was not used in any model training or in-sample validation and was only used to determine how models would respond to new data and to determine if the models overfitted to the training data.

We collected all search data in September of 2019 therefore all nowcasting done by developed models presented in this article will not exceed September 2019. All elastic net models were built and run in R version 3.6.2 using the *caret* and *glmnet* packages [40, 41]. Model fit was determined using Root Mean Square Error (RMSE), Mean Absolute Error (MAE), and $R^2$. All graphics of model fit, and search term correlation were created using *ggplot2* in R version 3.6.2, and search terms are presented as directly provided by Google Correlate[TM].

Elastic net regression is a penalized form of ordinary least squares regression and contains a hybrid of ridge and Least Absolute Shrinkage and Selection Operator (LASSO) regression penalties [29]. Elastic Net regression was implemented to both reduce the impact or outright eliminate non-essential feature data as it compromises the L1 and L2 penalties of LASSO and ridge regression respectively. Alpha and lambda hyper parameters are used in Elastic net regression to balance the tuning of the L1 (LASSO) and L2 (Ridge) norm penalty parameters (Eq 1A).

$$\text{Combined Penalty} \qquad \sum_{i=1}^{n} (Y - X\hat{\beta})^2 + \lambda \sum_{j=1}^{p} [(1-\alpha)(\hat{\beta}_j)^2 + \alpha|\hat{\beta}_j|] \qquad \text{Eq1A}$$

Alpha determines the relative weights of the two penalty parameters and lambda determines the overall weight of the summation of the individual penalties. For each region and elastic net model group (disease symptom and vector terms alone vs. full-term list), we tested a combination of 50 and 150 different automatically generated values of alpha and lambda to select optimal values.

Regional monthly Lyme disease incidence rates, as calculated from CDC surveillance data, was the outcome of the nowcasting models. Feature data was regional monthly search hit data from each region. We only used data from search terms where *gtrendsR* was able to appropriately return proportional monthly hit data. Despite terms having a correlation at the national level and therefore identified via Google Correlate[TM], some terms held non-variable values of zero for their monthly proportional hit data at the region level. These terms with their zero variance would cause model failure and thus were excluded from the modeling process.

### Variable importance

Elastic net regression can reduce or outright eliminate feature data from final models. We wanted to determine which search terms had the greatest influence in the final, best tuned models. To determine search terms influence, the *varImp* function from the caret package was used to calculate the scaled importance of each term in the final models. The *varImp* function takes the absolute value of each coefficient and ranks these coefficients and stores them as variable importance from zero to one hundred. Put simply, larger coefficients have greater influence and thus are associated with increased importance.

### Results

Between 2004 and 2017, the Northeast consistently had the highest counts of Lyme disease followed by the Midwest. The lowest incidence areas were consistently the West and Southwest regions (**Fig 2**). All regions showed seasonal oscillation of Lyme disease incidence with typical peaks in summer months (July, August, and September) and falling in winter (**Fig 3A**). Seasonal oscillation occurs at a lower incidence in the West and southwest regions compared to the high-incidence regions of the Northeast and Midwest (**Fig 3B and 3C**). These regional temporal trends were used with Google Correlate[TM] to identify 100 terms with correlated search patterns. Across all regions, there were environmental themes of outdoor activities that included concerts, camping, and water parks; places where people are likely to be exposed to

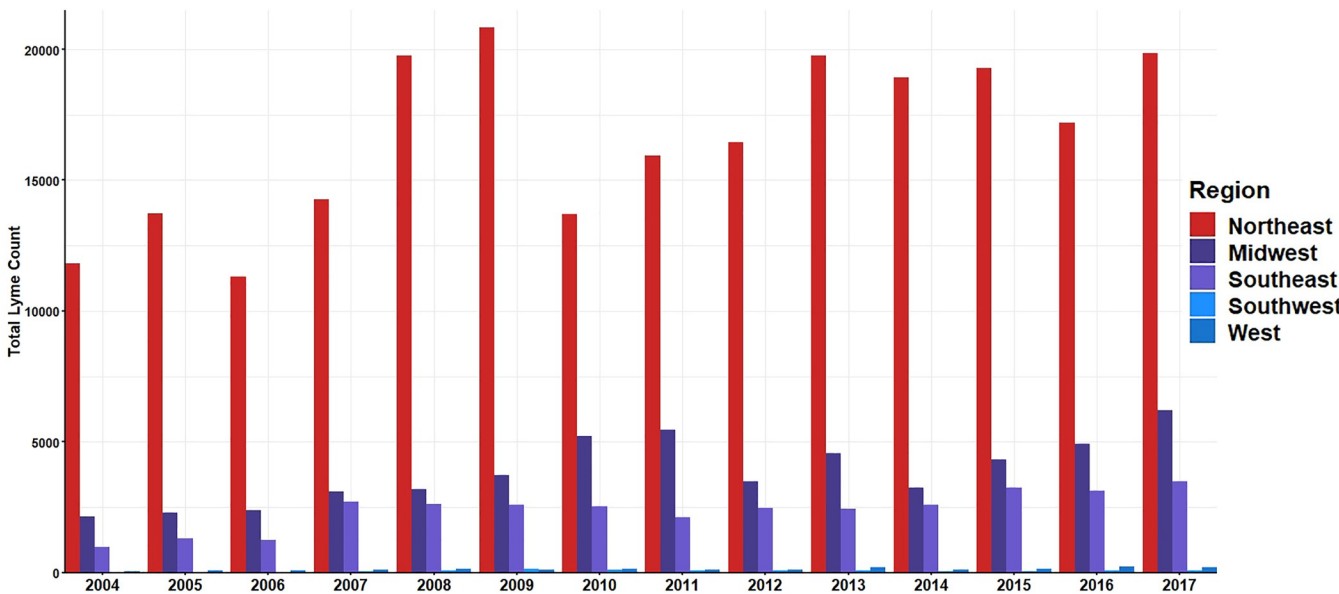

**Fig 2. Regional Lyme disease incidence count from CDC surveillance.** Incidence counts calculated by summing monthly state incidence form CDC surveillance in each region. Calculations and graphs made suing RStudio version 3.6.2.

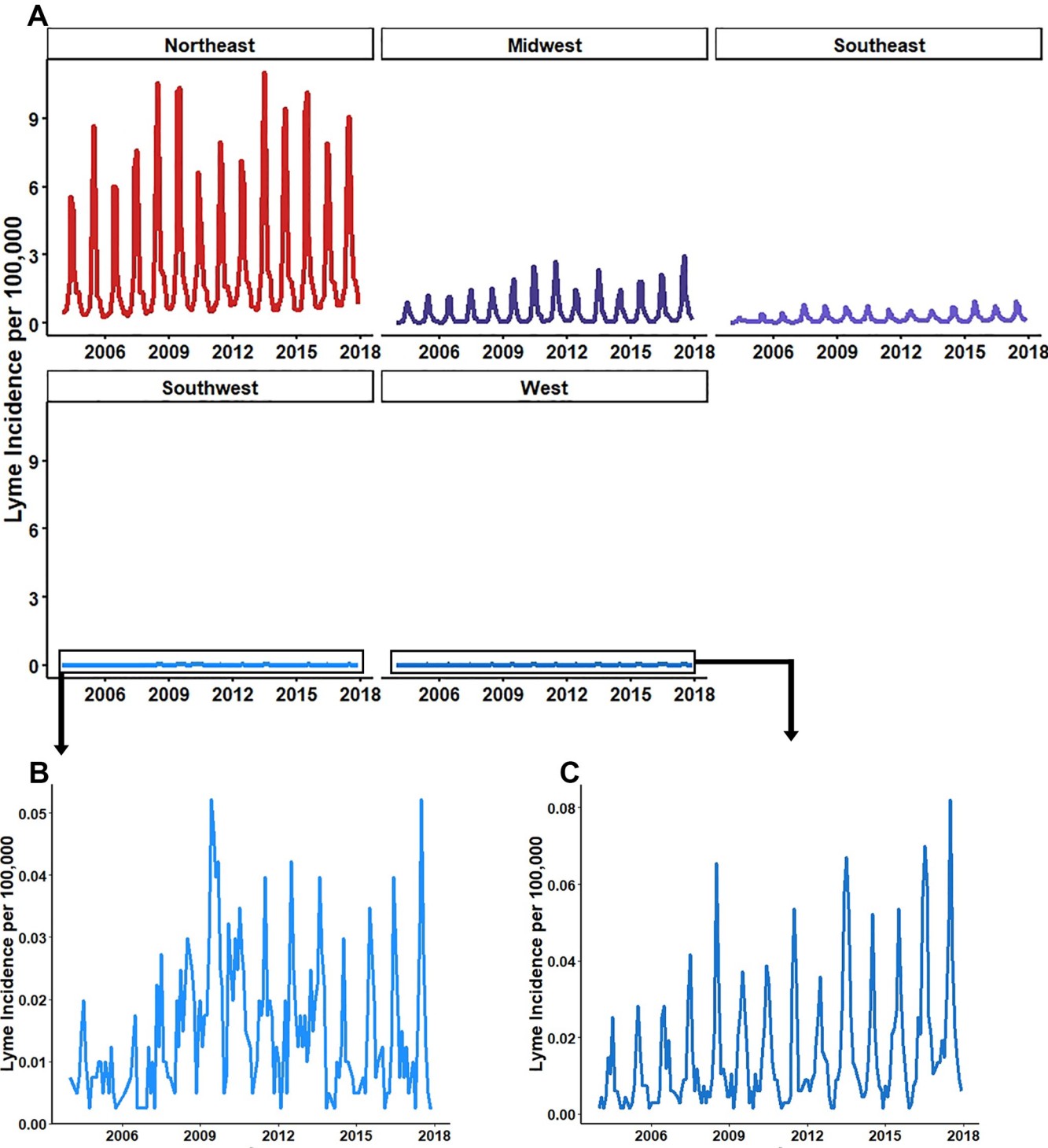

**Fig 3. Regional Lyme disease incidence. (A)** All regions relative to Northeast incidence rates, **(B)** Southwest, **(C)** West. Incidence rates calculated by summing monthly state incidence from CDC surveillance in each region. Denominator values calculated from 2010 US Census state populations and aggregated to region. Calculations and graphs made using RStudio version 3.6.2.

**Table 1. Candidate search terms identified via Google Correlate[TM] by region with symptom/vector terms.**

| Northeast Search Terms | Midwest Search Terms |
|---|---|
| Identified by Google Correlate[TM] | Identified by Google Correlate[TM] |
| free concerts, july calendar, necbl, little league all stars, alive at five, movies under the stars, prospect park bandshell, summer recipe, harwich mariners, freezer jam | festivals milwaukee, beaches in michigan, kings island discount, easy summer recipes, lake beaches, motel wisconsin dells, movies in the park, summer desserts, dorm bedding, drive in ohio |
| Southeast Search Terms | Southwest Search Terms |
| Identified by Goggle Correlate[TM] | Identified by Google Correlate[TM] |
| intex, cloudy pool, summer things, alabama water park, blue bayou in baton rouge, cloudy pool water, baking soda pool, summer things to do, green pool, springtails | loans for, how to make string bracelets, pigeon forge hotels, recipes on the grill, sandstone amphitheater, cheap bmx bikes, cataratas del niagara, world rv, cave of the winds colorado springs, produce stand |
| West Search Terms | Symptom and *Ixodid*-vector Terms |
| Identified by Google Correlate[TM] | added for Each Region |
| concert in the park, berry picking, movies in park, concert in park, blueberry picking, outdoor movies, soak city, lake water park, blueberry farm, broomfield bay | tick, black tick, lyme, lyme disease, rash, bullseye rash, bell's palsy, facial paralysis, side of face paralyzed, knee pain, swollen knees, swollen joint, swollen joints, joint pain, fever, tired, deer tick, black-legged tick, black legged tick, black leg tick |

Ixodes ticks during the late spring, summer and early fall [10]. (**Table 1,** complete list of candidate search terms provided in S1 Table). Gtrends was used to collect regional monthly proportional search data for each term identified from Google Correlate along with the symptom and vector terms (120 total terms for each region). Some terms identified with Google Correlate[TM] at the national level were identified as having no search traffic at the regional level and were removed from the regional list (**Table 2**).

For accurate modeling predictions, or nowcasts, it is important to use feature data that is correlated to the outcome data of interest. Pearson's correlation was performed for each term's proportional monthly search traffic and regional Lyme disease rates within the training timeframe. Individual term correlation with Lyme disease incidence had a large range for each region of the US with moderate mean and median correlation values (**Table 3,** complete results provided in S2 Table). The ten most correlated search terms for the training period were either strongly or moderately correlated with regional Lyme disease rates, except for terms matching the trend of rates, or lack thereof, in the Southwest (**Table 4**). Each region, except the Southwest, had sixteen terms with a correlation greater than 0.7 (complete results provided in S2 Table). Over the regions that have suitable *Ixodes* climate and habitat (Northeast, Midwest, Southeast, and West regions), we found high maximum correlation values (0.893, 0.898, 0.840, and 0.836, respectively) for the top correlated search terms. Many of the 21 terms based on known Lyme disease symptoms or vectors had poor bivariate correlation with regional Lyme disease incidence. For example, fever, which is more often searched in winter months [42], was negatively correlated with Lyme disease incidence in every region for the entire timeframe of the study (**Fig 4**).

The variance of feature data is also important for making accurate predictions. Features that have little to no variance overtime make for poor predictors. The variability of each term

**Table 2. Number of search terms that had monthly proportional hit data available from Gtrends[TM].**

| Region | Terms Into Gtrends[TM] | Terms From Gtrends[TM] |
|---|---|---|
| Northeast | 120 | 87 |
| Midwest | 120 | 86 |
| Southeast | 120 | 80 |
| Southwest | 120 | 42 |
| West | 120 | 83 |

**Table 3. Summary values of bivariate correlation of full-term list search terms to regional Lyme disease rates of model training data.**

| Region | Range | Mean Correlation | Median Correlation |
|---|---|---|---|
| Northeast | -0.279, 0.893 | 0.560 | 0.663 |
| Midwest | -0.245, 0.898 | 0.602 | 0.691 |
| Southeast | -0.137, 0.840 | 0.524 | 0.590 |
| Southwest | -0.065, 0.612 | 0.229 | 0.231 |
| West | -0.165, 0.836 | 0.421 | 0.416 |

was assessed per region. It was found that two terms in the Northeast, one term in the Midwest, and ten terms in the Southeast had zero variance. These terms were excluded from the nowcasting process.

To evaluate the hypothesis that nowcast predictions would be more accurate when including the full list of candidate search terms as compared to a list of Lyme disease specific terms, two sets of elastic net regression models were constructed: 1. models with only Lyme disease symptoms and vector terms as features and 2. models with the full list of non-zero variance terms identified from Google Correlate coupled with symptom and vector terms (S1 Table). Predictions from regression models developed using data from symptom and vector terms exclusively, produced accurate nowcasting models (assessed via $R^2$) with low error (assessed via RMSE and MAE) in four of the five US regions (**Table 5,** results for both models provided in S3 Table). The predictions from these models provide accurate estimations of the timing of the seasonal pattern of Lyme disease (**Fig 5**).

Nowcasting models developed using the full list of search terms produced predictions that had a 1.33-fold improvement in accuracy and a 0.5-fold reduction in error compared to the

**Table 4. Ten most correlated regional search terms for training period (2004–2012).**

| Northeast | | Midwest | | Southeast | | Southwest | | West | |
|---|---|---|---|---|---|---|---|---|---|
| Search Term | Corr. Value | Search Term | Corr. Value | Search Term | Corr. Value | Search Term | Corr. Value | Search Term | Corr. Value |
| july calendar | 0.89** | kings island discount | 0.90** | intex | 0.84** | loans for | 0.61** | movies in park | 0.84** |
| free concerts | 0.88** | beaches in michigan | 0.90** | cloudy pool | 0.84** | hotels ca | 0.55** | movies in the park | 0.83** |
| movies under the stars | 0.87** | festivals milwaukee | 0.89** | summer things | 0.81** | ca water | 0.55** | movie in park | 0.82** |
| lyme | 0.85** | easy summer recipes | 0.88** | baking soda pool | 0.80** | deer tick | 0.45* | concert in the park | 0.80** |
| summer recipe | 0.85** | lake beaches | 0.88** | green pool | 0.80** | moon bay ca | 0.44* | berry picking | 0.80** |
| lyme disease | 0.85** | motel wisconsin dells | 0.87** | alabama water park | 0.79** | half moon bay ca | 0.40* | blueberry farm | 0.79** |
| little league all stars | 0.85** | blueberry farm | 0.85** | cloudy pool water | 0.79** | make string bracelets | 0.40* | concert in park | 0.79** |
| necbl | 0.84** | summer desserts | 0.85** | summer things to do | 0.79** | rash | 0.39* | blueberry picking | 0.78** |
| berry picking | 0.83** | movies in the park | 0.85** | blue bayou in baton rouge | 0.77** | tick | 0.38* | outdoor movies | 0.77** |
| alive at five | 0.83** | watermelon recipe | 0.84** | springtails | 0.75** | how to make string bracelets | 0.38* | lake water park | 0.76** |

** p << 0.05

* p < 0.05.

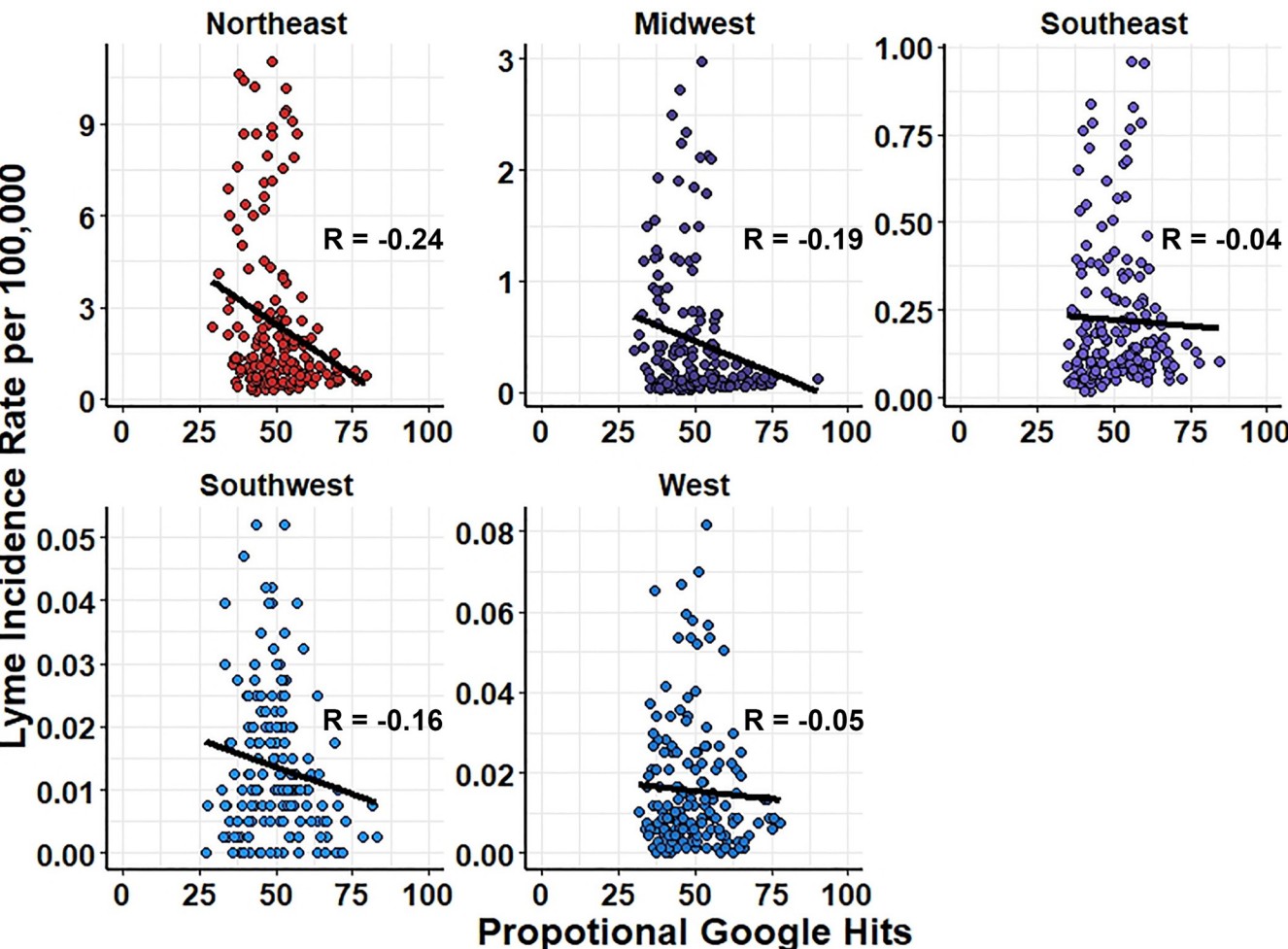

**Fig 4. Negative bivariate correlation of fever to Lyme disease incidence for all regions of the United States.** Correlation calculated using Pearson method with independent variable as proportional Google hits for each term and dependent variable Lyme Incidence per 100,000 for each region.

**Table 5. Predictions from symptoms and vector terms only models produce accurate predictions with low error.**

|  | Northeast | Midwest | Southeast | Southwest | West |
|---|---|---|---|---|---|
| $\alpha, \lambda$ | 0.47, 0.60 | 0.33, 0.20 | 0.29, 0.07 | 0.11, 0.01 | 0.1, 0.01 |
| **Training** |  |  |  |  |  |
| RMSE | 1.32 | 0.36 | 0.11 | 0.01 | 0.01 |
| MAE | 0.89 | 0.21 | 0.07 | 0.01 | 0.01 |
| $R^2$ | 0.77 | 0.65 | 0.67 | 0.32 | 0.50 |
| **In-sample Validation** |  |  |  |  |  |
| RMSE | 1.50 | 0.38 | 0.11 | 0.01 | 0.01 |
| MAE | 1.01 | 0.25 | 0.07 | 0.01 | 0.01 |
| $R^2$ | 0.71 | 0.59 | 0.69 | 0.38 | 0.29 |
| **Out of Sample** |  |  |  |  |  |
| RMSE | 1.65 | 0.43 | 0.14 | 0.01 | 0.01 |
| MAE | 1.38 | 0.34 | 0.10 | 0.01 | 0.01 |
| $R^2$ | 0.79 | 0.76 | 0.82 | 0.37 | 0.63 |

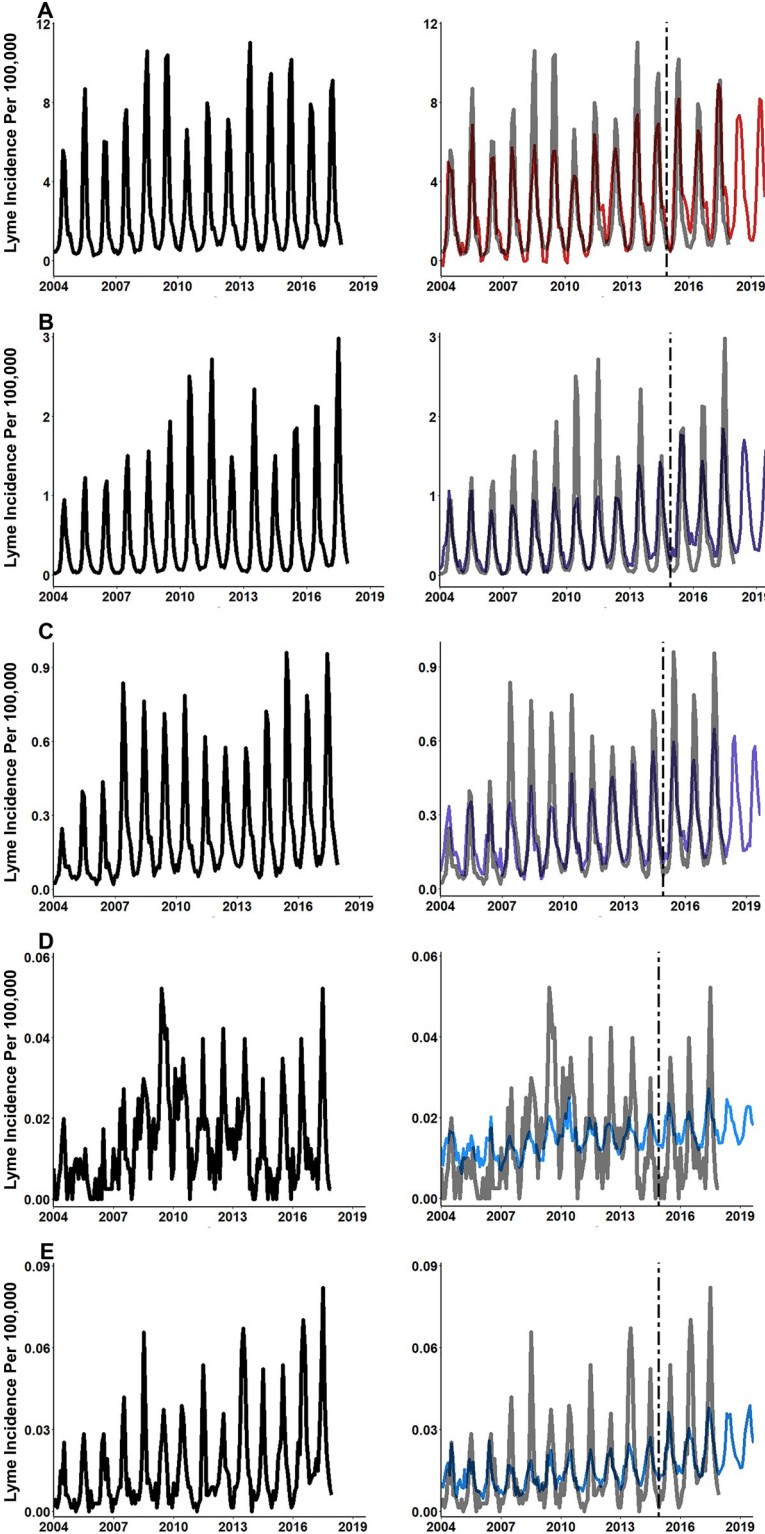

**Fig 5. Elastic net modeling using disease symptom and vector terms only produces accurate nowcasting model for Lyme disease.** (A) Northeast, (B) Midwest, (C) Southeast, (D) Southwest, and (E) West. Same one-year period from Northeast region with accurate nowcast model. Two elastic net models were developed for each region. Elastic net models trained using CDC surveillance data and search term data from February 2004 through December 2014. Vertical dashed line starts at January 2015 and indicates the start of the hold-out data set. Nowcasting performed using search term data from January 2018 until September 2019.

**Table 6. Predictions form full-term list models produce highly accurate predictions with low error.**

|  | Northeast | Midwest | Southeast | Southwest | West |
|---|---|---|---|---|---|
| **α, λ** | 0.1, 0.85 | 0.93, 0.00 | 0.1, 0.07 | 0.1, 0.01 | 0.1, 0.00 |
| **Training** | | | | | |
| **RMSE** | 0.66 | 0.12 | 0.06 | 0.01 | 0.01 |
| **MAE** | 0.46 | 0.09 | 0.04 | 0.01 | 0.00 |
| **$R^2$** | 0.94 | 0.95 | 0.91 | 0.56 | 0.84 |
| **In-sample Validation** | | | | | |
| **RMSE** | 0.99 | 0.23 | 0.08 | 0.01 | 0.01 |
| **MAE** | 0.62 | 0.14 | 0.05 | 0.01 | 0.01 |
| **$R^2$** | 0.87 | 0.85 | 0.84 | 0.44 | 0.70 |
| **Out of Sample** | | | | | |
| **RMSE** | 0.74 | 0.29 | 0.14 | 0.01 | 0.01 |
| **MAE** | 0.52 | 0.17 | 0.09 | 0.01 | 0.01 |
| **$R^2$** | 0.97 | 0.94 | 0.91 | 0.45 | 0.82 |

symptom and vector only models (**Table 6**, results for both models provided in S4 Table). For each region it was found that using the full-term lists, which often included environmentally themed terms, increased the accuracy and reduced the error of predictions. On average, model accuracy ($R^2$) improved by 0.2 when using the full list of search terms. The greatest improvement in accuracy when using the full-term list models was seen in the West ($R^2$ difference was 0.31). The Southeast had the least improvement (0.12) in accuracy. RMSE was reduced by 0.18 on average across all regions and MAE was reduced by 0.14 when comparing predictions between the full-term list models and the symptom and vector only models. The greatest reduction in error was seen in the Northeast region. It was found that predictions from the full-term list compared to the symptom and vector only models reduced RMSE by 0.69 and MAE by 0.56 cases per 100,000 population in the Northeast region. Reduction in error for the Southwest and West were found to be approximately 0 (RMSE = 0.001 and 0.004 respectively; MAE = 0.001 and 0.002 respectively). Predictions from the full list models also produced accurate timing of seasonal patterns of Lyme disease, but with improved mimicking of peaks and recessions (**Fig 6**). Compared to the symptom and vector term only models, predictions from the full-term list model showed more accurate variation in the spring and summer peaks of Lyme disease across all regions. In both modeling efforts, the Southwest consistently had the poorest predictive accuracy.

In some years, Lyme disease incidence in the Northeast and Midwest showed secondary peaks or plateaus in the post-summer spike of incident cases. These secondary spikes or plateaus typically occur in late summer and early fall months as infected adult ticks take blood meals transmitting Lyme disease to people. Predictions from models using only symptoms and vector terms did not have sufficient sensitivity to detect to these changes (**Fig 7A and 7C**). Alternatively, predictions from the full-term list models had sufficient sensitivity to detect these secondary spikes or plateaus of decreasing incidence at the regional level (**Fig 7B and 7D**).

Statistical learning techniques can help highlight specific areas in which future hypothesis or interventions could be generated. We identified the three most important terms from the accurate full-term list nowcasting models. (**Table 7**). As hypothesized, many of the top three most important terms for producing accurate nowcasts were regionally specific and environmentally themed. The Northeast and Southeast were the only regions that had a potential symptom term (bulls-eye rash, rash) identified in the top three important terms. We further

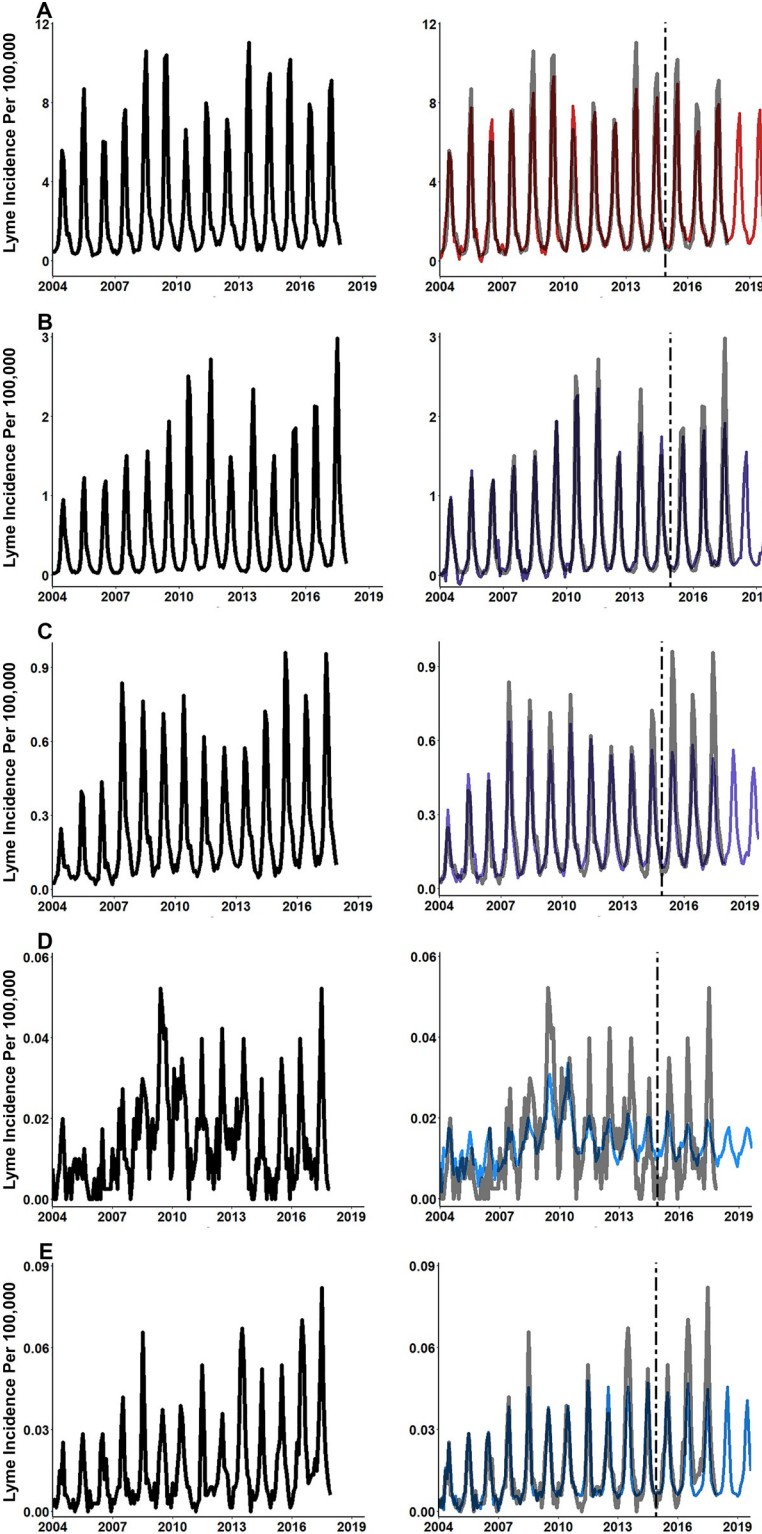

**Fig 6. Elastic net modeling using the full-term list produces predictions with greater accuracy and less error. (A)** Northeast, **(B)** Midwest, **(C)** Southeast, **(D)** Southwest, and **(E)** West. Same one-year period from Northeast region with accurate nowcast model. Two elastic net models were developed for each region. Elastic net models trained using CDC surveillance data and search term data from February 2004 through December 2014. Vertical dashed line starts at January 2015 and indicates the start of the hold-out data set. Nowcasting performed using search term data from January 2018 until September 2019.

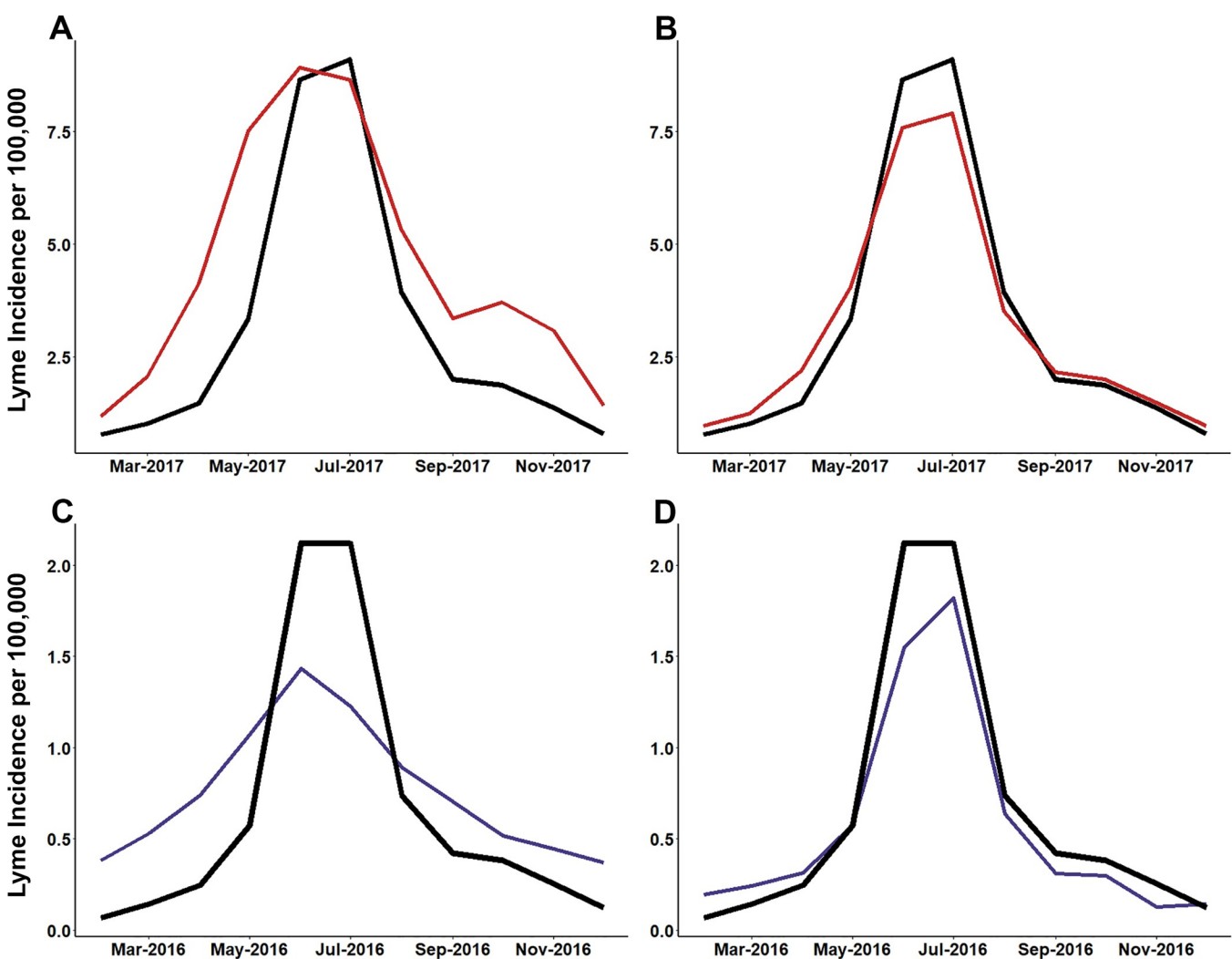

**Fig 7. Elastic net modeling using full-term list is sensitive to secondary spikes of Lyme disease incidence in Northeast and Midwest regions. (A)** Northeast Lyme disease incidence (black line) and disease symptom and vector terms only model predictions (red line), **(B)** Northeast Lyme disease incidence (black line) and full-term list model predictions (red line), **(C)** Midwest Lyme disease incidence (black line) and disease symptom and vector terms only model predictions (purple line), and **(D)** Midwest Lyme disease incidence (black line) and full-term list model predictions (purple line). Elastic net models trained using CDC surveillance data and search term data from February 2004 through December 2014 and hold-out data from January 2015 and December 2017.

hypothesized that due to the importance of these environmentally related themes, the time series of these search terms trends would mimic the same general trends for Lyme disease. These patterns are particular evident in areas with higher incidence of Lyme disease; the Northeast, Midwest and Southeast (**Fig 8**). It was found that the search traffic for these top three terms aligns with the peaks and recessions of Lyme disease on the same monthly scale.

## Discussion

With the growing incidence of Lyme disease in the United States, novel methods that help health departments to prepare for years of increased Lyme disease exposure are critical. We found that when using Google search history data in nowcasting, accurate predictions of Lyme disease can be generated. Importantly, the search traffic for the top three search terms generally followed the same temporal nature of regional Lyme disease incidence. These terms and

**Table 7. Three most important terms for each model often environmentally themed.**

| Northeast | | | |
|---|---|---|---|
| Elastic Net 1 | | Elastic Net 2 | |
| Search Term | Scaled Importance | Search Term | Scaled Importance |
| July Calendar | 100.00 | July Calendar | 100.00 |
| Fresh Cherry Pie | 82.12 | Fresh Cherry Pie | 83.29 |
| Bullseye Rash | 75.51 | Bullseye Rash | 75.47 |
| **Midwest** | | | |
| Elastic Net 1 | | Elastic Net 2 | |
| Search Term | Scaled Importance | Search Term | Scaled Importance |
| Festivals Milwaukee | 100.00 | Festivals Milwaukee | 100.00 |
| Lake Beaches | 97.35 | Kings Island Discount | 99.16 |
| Kings Island Discount | 96.35 | Lake Beaches | 97.40 |
| **Southeast** | | | |
| Elastic Net 1 | | Elastic Net 2 | |
| Search Term | Scaled Importance | Search Term | Scaled Importance |
| Intex Pool Cover | 100.00 | Intex Pool Cover | 100.00 |
| Rash | 87.07 | Rash | 88.06 |
| Swampdogs | 85.64 | Swampdogs | 85.45 |
| **Southwest** | | | |
| Elastic Net 1 | | Elastic Net 2 | |
| Search Term | Scaled Importance | Search Term | Scaled Importance |
| Loans for | 100.00 | Loans for | 100.00 |
| CA Water | 67.20 | CA Water | 66.82 |
| Hotels CA | 61.00 | Hotels CA | 60.14 |
| **West** | | | |
| Elastic Net 1 | | Elastic Net 2 | |
| Search Term | Scaled Importance | Search Term | Scaled Importance |
| Movies in the Park | 100.00 | Movies in the Park | 100.00 |
| Concert in the Park | 69.18 | Concert in the Park | 69.65 |
| Waterworld Denver | 62.13 | Waterworld Denver | 62.44 |

nowcasting methods could help Health Departments determine approximate trends of Lyme disease in their area by monitoring the search traffic trends of the terms via the free tool of Google Trends[TM]. Additionally, many of the terms that remained in these accurate models were environmentally themed and can be used to generate future hypotheses for intervention and prevention actives.

Overall, each elastic net model performed well and provided accurate estimations of the of regional Lyme disease incidence provided by surveillance data from the CDC (**Tables 5** and **6**). Results showed that predictions were more accurate from models using a full list of colloquial search terms the average person is likely to search compared to models that only used symptom, disease or vector terms. It was also found that predictions from models that included the full-term list were more sensitive to detecting secondary spikes and recession plateaus in the fall months of the Northeast and Midwest (**Fig 7**). Moreover, many of the search terms identified via Google Correlate which had high levels of bivariate correlation and remained important throughout the elastic net modeling process were environmentally related. While not all these terms directly relate to an activity that have obvious risk of tick exposure and transmission of Lyme disease, environmentally related terms can serve as a proxy for an intention for people to spend time outdoors. Increased time spent outdoors has

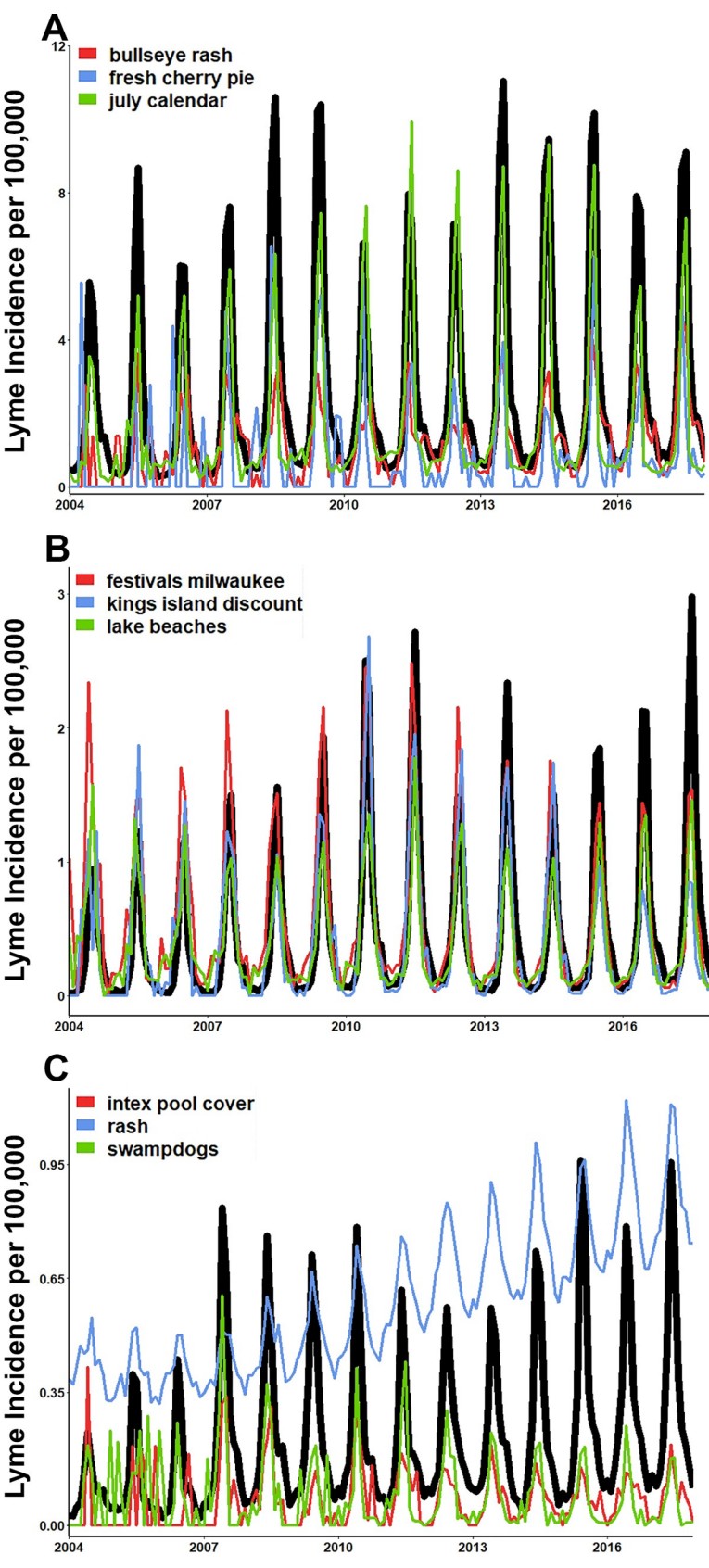

**Fig 8. Time series of regional candidate search terms for simple Lyme disease tracking. (A).** Northeast, **(B).** Midwest, and **(C).** Southeast. The top three most important terms from each region model identified by *varImp* function in R. (a-c). Candidate terms scaled to align with regional Lyme disease incidence. Terms presented directly as provided by Google Corrleate[TM].

been shown to increase exposure to ticks in the environment [43–45]. Causal inference cannot be directly drawn from these results, however given the common pattern of environmental terms and many of their high correlations a pattern has emerged. These terms can help LHDs generate hypothesis on where to perform future tick surveillance, implement intervention measures, or spread tick awareness. These findings suggest the importance of including collo-quial search terms over symptom or vector related terms alone for current and future predic-tion efforts. Our models can be implemented by LHDs as they currently are, or terms that more specific the local populations search habits can be substituted to further improve performance.

The Southwest, a non-endemic region for Lyme disease [14], continually had the poorest performing predictions. *Ixodes* ticks in the Southwest are more suspected to feed on lizards and other non-reservoir hosts [46], thus it is not surprising that Lyme disease incidence was low. The CDC also classifies county of residence and not county of acquisition in surveillance reports therefore it is likely that those diagnosed in this region were exposed elsewhere. The Southwest also had the lowest number of feature data compared to all other regions. These all likely led to the low performance of predictions in this region. On the other hand, the West region, which also had a low number of incident cases, but had a greater number of feature data had better performing model predictions. The West also has suitable habitat for *Ixodes pacificus*, a known vector of *B. burgdorferi* [14]. These results indicate that in addition to hav-ing an appropriate number of feature data and outcomes, regions also need to have a suitable environment for the tick vectors in order to produce accurate nowcasts. These findings con-tinue to show the importance of inducing environment related feature data for current or future prediction efforts in areas that are either endemic with Lyme disease or have suitable *Ixodid* tick habitats.

To our knowledge, two prior studies have been performed using Google search data to try and improve model performance [47, 48]. One study concluded that using a single term, "Bor-reliose", was not helpful in improving model accuracy [47]. While "Borreliose" is a medically accurate term for Lyme disease, we found that colloquial disease terms had moderate to high levels of correlation. Our findings found that the bivariate correlation for disease symptoms and colloquial disease terms ranged from -0.33 to 0.85 across five US regions. Terms often moderately (correlation value > 0.5), or highly correlated (correlation value > 0.8), with regional monthly Lyme disease incidence included: "lyme disease", "lyme", "rash", and "tick". Further, environmentally related terms often had the highest levels of correlation across all regions. Another study developed a tool, Lymelight, which monitored the incidence of Lyme disease in real time using Lyme disease symptom web searches in a two-year period to predict future Lyme disease burden and treatment impacts [48]. Despite producing accurate models, this method only used symptom terms which may not predict true patterns of Lyme disease or risky behaviors. Our findings show using symptom, disease and vector terms in combination with terms that focus on environments in which one may have the risk of being exposed can greatly improve model performance over symptom and vector terms alone. These findings continue to suggest the importance of direct or proxy measures for time spent outdoors when predicting vector-borne diseases.

An advantage of using data from Google search history, R studio as a modeling software, and elastic net regression is that accurate predictions can be made quickly (approximately 24

hours from start to finish) and free. This can allow LHDs to have more up to date estimations of regional Lyme disease incidence beyond federal report schedules without additional finical burden. We found when graphing the search traffic for three most important terms from regional models, in endemic areas of the Northeast and Midwest, as hypothesized they provide a very good broad scale of timing. Following these terms, or more locally specific environmental terms could provide even quicker tracking of general temporal trends of Lyme disease for LHDs. Most of the top three important terms were environmentally related. This further suggests the importance of including terms or variables that focus on the environment for current and future prediction efforts.

While there are strengths of statistical learning approaches, there are limitations to our approach as well. These models were developed at the regional level and are subject to less accurate predictions at the state or local level without refitting the model. Additionally, grouping states into different regions will alter results of these findings as both regional rate and search term identification using Google Correlate™ were performed regional aggregation strategy. These models are not generalizable to other vector-borne diseases in their current form. Similar approaches could be used for other vector-borne diseases such as Anaplasmosis, as this is also vectored by *Ixodid* ticks and therefore will have similar temporal trends and environmental risk factors. Additionally, these models are not generalizable to other countries. All the Lyme disease and search data were based on US disease and Google habits, it is unlikely that our developed models would produce accurate results in other countries. However, a similar approach could be used in other countries that have strong surveillance data and a free access database of the countries' most utilized search engine. Moreover, other sources of data on human behavior (i.e. data form social networks like Twitter) present additional opportunities for such models, potentially at greater spatial and temporal granularity. Greater consideration or different modeling techniques may need to be implemented for communicable diseases. However, these models can be incorporated to get a general idea of surrounding areas for those LHDs that are vastly underfunded. Local or regionally specific terms could easily be substituted into these models which could help improve model fit on a case-by-case basis. These findings highlight the importance of strong disease surveillance and computational modeling efforts working together. Predictions over time are likely to improve not only due to increases in statistical and computing power, but in the maintenance and enhancement of strong disease surveillance efforts performed nationwide.

## Supporting information

**S1 Table. Complete list of search terms identified by Google Correlate from each region.** Terms for each region were identified via Google Correlate using region specific Lyme disease rates from training period data.
(PDF)

**S2 Table. Bivariate correlations of each search term to the regional Lyme disease rates.** Pearson Correlations values were calculated between each term monthly proportional search data and corresponding Lyme disease rates for each term and region.
(PDF)

**S3 Table. Predictions from symptoms and vector terms only models produce accurate predictions with low error.**
(PDF)

**S4 Table. Predictions from full list models produce highly accurate predictions with low error.**
(PDF)

## Author Contributions

**Conceptualization:** Eric Kontowicz, Grant Brown, James Torner, Margaret Carrel, Kelly K. Baker, Christine A. Petersen.

**Data curation:** Eric Kontowicz.

**Formal analysis:** Eric Kontowicz.

**Supervision:** Christine A. Petersen.

**Visualization:** Eric Kontowicz.

**Writing – original draft:** Eric Kontowicz.

**Writing – review & editing:** Eric Kontowicz, Grant Brown, James Torner, Margaret Carrel, Kelly K. Baker, Christine A. Petersen.

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
