## [Decision Letter · Decision Letter 0]

18 Jun 2021

PONE-D-21-11338

Inclusion of environmentally themed search terms improved Elastic Net regression nowcasts of regional Lyme disease rates

PLOS ONE

Dear Christy:

Thank you for submitting your manuscript to PLOS ONE. After careful consideration, we feel that it has merit but does not fully meet PLOS ONE’s publication criteria as it currently stands. Therefore, we invite you to submit a revised version of the manuscript that addresses the points raised during the review process.

One of the reviewers does not think that infodemiology has any utility and rejected it.  I had to find reviewers who had actually done this kind of work before.  One of them had substantive comments that need to be addressed in a revision.  In comments to me, it was suggested that the authors needed more familiarity with infodemiology, particularly GoogleTrends analyses, and that there was a good literature on its applications, limitations, and standards of practice.  The other comment was that the database was at least 2 years old and it was likely that there was additional data that could be used.  I

We look forward to receiving your revised manuscript.

Best regards,

Sam

Sam R. Telford III

Academic Editor

PLOS ONE

Journal Requirements:

2. Please confirm in your manuscript that you have adhered to the Terms and Use/Terms of Service of Google Correlate.

"The funders had no role in study design, data collection and analysis, decision to

publish, or preparation of the manuscript."

Reviewers' comments:

Reviewer's Responses to Questions

**Comments to the Author**

1. Is the manuscript technically sound, and do the data support the conclusions?

Reviewer #1: No

Reviewer #2: Yes

Reviewer #3: Yes

2. Has the statistical analysis been performed appropriately and rigorously? 

Reviewer #1: Yes

Reviewer #2: Yes

Reviewer #3: Yes

3. Have the authors made all data underlying the findings in their manuscript fully available?

Reviewer #1: Yes

Reviewer #2: Yes

Reviewer #3: No

4. Is the manuscript presented in an intelligible fashion and written in standard English?

Reviewer #1: Yes

Reviewer #2: Yes

Reviewer #3: Yes

5. Review Comments to the Author

Reviewer #1: This manuscript proposes a method to forecast Lyme disease incidence using regressions methods with Google search history data. I believe that the application of machine learning/Big Data methods is misguided for ecologically complex phenomena like the incidence tick-borne diseases. This manuscript does not provide any compelling evidence for the contribution of these techniques to the understand-ing of Lyme disease incidence. As an exquisitely seasonal process, Lyme disease perpetuation and zo-onotic transmission will be powerfully correlated to Google search terms. The same would be true for indicators of “nice weather” (not obtained from Google!).

Reviewer #2: General comments

This is a paper on inclusion of environmentally themed search terms improved Elastic Net regression nowcasts of regional Lyme disease rates. I have some comments on your manuscript.

Specific comments

1. Abstract and Introduction: ”…with 95% of human cases occurring…” Please provide the absolute numbers of cases (n/N), showing where this percentage is coming from.

2. Introduction: “CDC”. Please write this out when mentioned for the first time in the main text.

3. Introduction: “LASSO”. Please write this out when mentioned for the first in the text.

4. Material and Methods: “erythema migrans”. You may consider briefly describing what this is, you may put this description in the parentheses, for example.

5. Please check up the capital letters concerning Google, United States, Table, for example. Also, some words are lowercased instead of capital letters.

6. Both abbreviations are used: “US” and U.S.”. Please consider choosing one of them.

7. Figure 7: If you use color lines in the figures, please tell the readers which color indicates which line.

8. Please check up the reference list concerning the links and make sure that they are updated.

Reviewer #3: This is an interesting approach in modeling Lyme Disease with Google Trends data. However, there are some issues that need to be addressed before this manuscript can be reconsidered for publication.

The authors mention that “High correlation was determined when the correlation value was greater than 0.8, moderate if correlation value was between 0.5 and 0.8, and poor when less than 0.5”. Shouldn’t the significance of a correlation be measured by, for example, the p-values (or CIs)? Also, “high” and “moderate” should be defined (I assume the authors mean that high is p<0.01 and that moderate is p<0.05; however, a correlation with a p-value less than .05 is considered quite strong).

There is no description of the Google Trends data selection criteria and collection procedure. This is an important drawback of this manuscript. All methodology steps should be reported in detail (e.g., period, region, category, web search, use of quotes for keywords with more than one word, individual searches, comparisons, etc.).

This is an information epidemiology (infodemiology) study. I suggest that the authors study the relevant literature in order to gain insight and enhance their literature review. An introductory paragraph could be added in the Introduction Section.

The analysis (data collection) was conducted in September 2019, considering data up to December 2018. It is now 2021, and there are two more years’ data available. I believe it would add to the value of this manuscript if the analysis was updated.

6. PLOS authors have the option to publish the peer review history of their article (what does this mean?). If published, this will include your full peer review and any attached files.

Reviewer #1: **Yes: **Ivo M Foppa

Reviewer #2: **Yes: **Samuli Pesälä

Reviewer #3: No

---

## [Author Response · Author response to Decision Letter 0]

24 Aug 2021

We thank our reviewers for their thoughtful comments. By responding to them, we believe that we have greatly strengthened the manuscript. Our responses will appear below in Garamond.

 There were no sources of funding for this research

 The funders had no role in study design, data collection and analysis, decision to

publish, or preparation of the manuscript.

 No authors received a salary from any of our funders.

 The authors received no specific funding for this work.

5. Review Comments to the Author

Reviewer #1: This manuscript proposes a method to forecast Lyme disease incidence using regressions methods with Google search history data. I believe that the application of machine learning/Big Data methods is misguided for ecologically complex phenomena like the incidence tick-borne diseases. This manuscript does not provide any compelling evidence for the contribution of these techniques to the understanding of Lyme disease incidence. As an exquisitely seasonal process, Lyme disease perpetuation and zoonotic transmission will be powerfully correlated to Google search terms. The same would be true for indicators of “nice weather” (not obtained from Google!).

We appreciate the considerations of this reviewer and completely agree that the ecological system of Borrelia is complex. This exercise was to determine how well these computational methods could reproduce and “nowcast” trends of reported Lyme Disease, and surprisingly it did quite well and found trends even if the weather was not nice… like the fall adult Ixodes transmission “bump”. That is why we feel this should be reported through this work.

Reviewer #2: General comments

This is a paper on inclusion of environmentally themed search terms improved Elastic Net regression nowcasts of regional Lyme disease rates. I have some comments on your manuscript.

Specific comments

1. Abstract and Introduction: ”…with 95% of human cases occurring…” Please provide the absolute numbers of cases (n/N), showing where this percentage is coming from.

We have now included the total number of confirmed positive cases (n) from northeast and upper Midwest states and the total number of confirmed positive cases for the United

states (N).

2. Introduction: “CDC”. Please write this out when mentioned for the first time in the main text.

We have included the full name (Center for Disease Control and Prevention) of the CDC

before its first mention in the introduction

3. Introduction: “LASSO”. Please write this out when mentioned for the first in the text.

We have included the full name (Least Absolute Shrinkage and Selection Operator) of

the LASSO before its first mention in the introduction.

4. Material and Methods: “erythema migrans”. You may consider briefly describing what this is, you may put this description in the parentheses, for example.

We have added “bullseye rash” as a parenthetical comment in this section.

5. Please check up the capital letters concerning Google, United States, Table, for example. Also, some words are lowercased instead of capital letters.

We have made edits throughout the manuscript to maintain consistency of capitalization and apologize for these errors.

6. Both abbreviations are used: “US” and U.S.”. Please consider choosing one of them.

We have edited the manuscript to maintain consistency of US throughout. 

7. Figure 7: If you use color lines in the figures, please tell the readers which color indicates which line.

We have included text into the legend of Figure 7 to make clear that the black lines are regional incidence and the colored lines are model predictions. 

8. Please check up the reference list concerning the links and make sure that they are updated.

We have updated the reference list to ensure that the all links are active and

functioning.

Reviewer #3: This is an interesting approach in modeling Lyme Disease with Google Trends data. However, there are some issues that need to be addressed before this manuscript can be reconsidered for publication.

1. The authors mention that “High correlation was determined when the correlation value was greater than 0.8, moderate if correlation value was between 0.5 and 0.8, and poor when less than 0.5”. Shouldn’t the significance of a correlation be measured by, for example, the p-values (or CIs)? Also, “high” and “moderate” should be defined (I assume the authors mean that high is p<0.01 and that moderate is p<0.05; however, a correlation with a p-value less than .05 is considered quite strong).

We thank the reviewer for this comment, and indicated when p-values were significant

in Supplemental Table 2 to accompany the correlation values. We feel that reporting the

correlation value directly is important, as this measures the strength of the linear

relationship directly. P-values (and by extension confidence intervals) measure the

strength of evidence for the presence of nonzero correlation, but do not give any

indication of the strength of correlation itself. In a large sample with multiple comparisons, one could obtain very strong evidence for nonzero correlation when the linear relationship is actually quite weak, while in a small sample a high correlation might not reach significance.

2. There is no description of the Google Trends data selection criteria and collection procedure. This is an important drawback of this manuscript. All methodology steps should be reported in detail (e.g., period, region, category, web search, use of quotes for keywords with more than one word, individual searches, comparisons, etc.).

We have added to the methods section to make this clear. We state in the methods that we use the terms identified via Google Correlate to collect search hit data on. We have added more language to make it clear that terms identified from Google Correlate were inputted into Google Trends unaltered to collect search term hit data for each region. I also included text to make it clear that gtrendsR is an R interface for Google Trends that allows for an automated process of collecting search term data. 

3. This is an information epidemiology (infodemiology) study. I suggest that the authors study the relevant literature in order to gain insight and enhance their literature review. An introductory paragraph could be added in the Introduction Section.

We have included a paragraph, lines 93-102, to the introduction outlining infodemiology and its use in predicting disease and better informing the general public about health-related outcomes. 

4. The analysis (data collection) was conducted in September 2019, considering data up to December 2018. It is now 2021, and there are two more years’ data available. I believe it would add to the value of this manuscript if the analysis was updated.

We appreciate and understand the reviewers concern for having recent data for publication. However, the authors’ intention of this work is to show that value of including environmentally related features when nowcasting with Google Search terms, which does not require all recent data. This manuscript highlights the importance of considering environmental factors when creating prediction models for vector borne diseases. To this end, the models have also been posted on Eric Kontowicz’s Github (https://github.com/ekontowicz/Lyme-disease-Elastic-Net-regression-Nowcasting) for further use by other researchers and additional updates. Lastly, given the time it takes to have rigorous peer review, particularly during this COVID-19 pandemic these findings will lag all surveillance data available.

---

## [Decision Letter · Decision Letter 1]

29 Sep 2021

PONE-D-21-11338R1Inclusion of environmentally themed search terms improved Elastic Net regression nowcasts of regional Lyme disease ratesPLOS ONE

Dear Christy,

Thank you for submitting your manuscript to PLOS ONE. After careful consideration, we feel that it has merit but does not fully meet PLOS ONE’s publication criteria as it currently stands. Therefore, we invite you to submit a revised version of the manuscript that addresses the points raised during the review process.

We look forward to receiving your revised manuscript.

Kind regards,

Sam R. Telford III

Academic Editor

PLOS ONE

Journal Requirements:

Additional Editor Comments (if provided):

Reviewer 2 makes the point that these kinds of studies should follow a standard methodology, e.g., Mavragani and Ochoa 2019 JMIR Public Health and Surveillance. I have examined this reference and believe that if you have a table in the methods or at lines 380 et seq that summarize the 4 critical aspects of this kind of work, as recommended by Mavragani and Ochoa (keywords, region, period, category) this concern would be satisfied. Citing this reference as informing your study reinforces the fact that standardized methods should be the basis for analyses using GT.   At the very least, because it is suggested that GT might be used by local health departments in a predictive manner, it might be good to make it easy for them to test it out by having a very simple set of search terms provided. It is clear that you spent much time analyzing keywords; in the end, what were they? (Table S1 is very comprehensive but can the most high-value keywords be highlighted in a summary table?) For region, was it overall, including metropolitan, urban, suburban and rural? For period, it looks like the specific searches were monthly from 2004-2015. According to Mavragani and Ochoa, search term category does not need to be specified if the keywords are very specific and you provided much analysis on selecting the useful keywords.

I would prefer not to send this back to the one reviewer and delay a decision. If you can provide such a table, the ms could be accepted without a third round of review.

Reviewers' comments:

Reviewer's Responses to Questions

**Comments to the Author**

1. If the authors have adequately addressed your comments raised in a previous round of review and you feel that this manuscript is now acceptable for publication, you may indicate that here to bypass the “Comments to the Author” section, enter your conflict of interest statement in the “Confidential to Editor” section, and submit your "Accept" recommendation.

Reviewer #2: All comments have been addressed

Reviewer #3: (No Response)

2. Is the manuscript technically sound, and do the data support the conclusions?

Reviewer #2: Yes

Reviewer #3: Yes

3. Has the statistical analysis been performed appropriately and rigorously? 

Reviewer #2: Yes

Reviewer #3: Yes

4. Have the authors made all data underlying the findings in their manuscript fully available?

Reviewer #2: Yes

Reviewer #3: Yes

5. Is the manuscript presented in an intelligible fashion and written in standard English?

Reviewer #2: Yes

Reviewer #3: Yes

6. Review Comments to the Author

Reviewer #2: Authors have revised the text as requested. This manuscript has definitely improved after revisions.

Reviewer #3: The GT methodology is still not properly reported. Please see relevant representative literature to understand how to report the methodology (like this one for example: https://www.jmir.org/2020/8/e19611/).

7. PLOS authors have the option to publish the peer review history of their article (what does this mean?). If published, this will include your full peer review and any attached files.

Reviewer #2: **Yes: **Samuli Pesälä

Reviewer #3: No

---

## [Author Response · Author response to Decision Letter 1]

21 Nov 2021

We thank Dr. Telford, the editor, and our reviewers for their thoughtful comments. We agree that the rigorous context of Mavragani and Ochoa, 2019, and in a new summary table, containing what they emphasize as the four key components for this type of computational epidemiology adds great strength to this method and specifically our manuscript. We have added this summary table at line 293 containing the five regions, the top 10, focused, keywords (so categories not needed) over the critical training period, 2004-2012. 

Our responses will appear below in Garamond.

 There were no sources of funding for this research

 The funders had no role in study design, data collection and analysis, decision to

publish, or preparation of the manuscript.

 No authors received a salary from any of our funders.

 The authors received no specific funding for this work.

5. Review Comments to the Author

Reviewer #1: This manuscript proposes a method to forecast Lyme disease incidence using regressions methods with Google search history data. I believe that the application of machine learning/Big Data methods is misguided for ecologically complex phenomena like the incidence tick-borne diseases. This manuscript does not provide any compelling evidence for the contribution of these techniques to the understanding of Lyme disease incidence. As an exquisitely seasonal process, Lyme disease perpetuation and zoonotic transmission will be powerfully correlated to Google search terms. The same would be true for indicators of “nice weather” (not obtained from Google!).

We appreciate the considerations of this reviewer and completely agree that the ecological system of Borrelia is complex. This exercise was to determine how well these computational methods could reproduce and “nowcast” trends of reported Lyme Disease, and surprisingly it did quite well and found trends even if the weather was not nice… like the fall adult Ixodes transmission “bump”. That is why we feel this should be reported through this work.

Reviewer #2: General comments

This is a paper on inclusion of environmentally themed search terms improved Elastic Net regression nowcasts of regional Lyme disease rates. I have some comments on your manuscript.

Specific comments

1. Abstract and Introduction: ”…with 95% of human cases occurring…” Please provide the absolute numbers of cases (n/N), showing where this percentage is coming from.

We have now included the total number of confirmed positive cases (n) from northeast and upper Midwest states and the total number of confirmed positive cases for the United

states (N).

2. Introduction: “CDC”. Please write this out when mentioned for the first time in the main text.

We have included the full name (Center for Disease Control and Prevention) of the CDC

before its first mention in the introduction

3. Introduction: “LASSO”. Please write this out when mentioned for the first in the text.

We have included the full name (Least Absolute Shrinkage and Selection Operator) of

the LASSO before its first mention in the introduction.

4. Material and Methods: “erythema migrans”. You may consider briefly describing what this is, you may put this description in the parentheses, for example.

We have added “bullseye rash” as a parenthetical comment in this section.

5. Please check up the capital letters concerning Google, United States, Table, for example. Also, some words are lowercased instead of capital letters.

We have made edits throughout the manuscript to maintain consistency of capitalization and apologize for these errors.

6. Both abbreviations are used: “US” and U.S.”. Please consider choosing one of them.

We have edited the manuscript to maintain consistency of US throughout. 

7. Figure 7: If you use color lines in the figures, please tell the readers which color indicates which line.

We have included text into the legend of Figure 7 to make clear that the black lines are regional incidence and the colored lines are model predictions. 

8. Please check up the reference list concerning the links and make sure that they are updated.

We have updated the reference list to ensure that the all links are active and

functioning.

Reviewer #3: This is an interesting approach in modeling Lyme Disease with Google Trends data. However, there are some issues that need to be addressed before this manuscript can be reconsidered for publication.

1. The authors mention that “High correlation was determined when the correlation value was greater than 0.8, moderate if correlation value was between 0.5 and 0.8, and poor when less than 0.5”. Shouldn’t the significance of a correlation be measured by, for example, the p-values (or CIs)? Also, “high” and “moderate” should be defined (I assume the authors mean that high is p<0.01 and that moderate is p<0.05; however, a correlation with a p-value less than .05 is considered quite strong).

We thank the reviewer for this comment, and indicated when p-values were significant

in Supplemental Table 2 to accompany the correlation values. We feel that reporting the

correlation value directly is important, as this measures the strength of the linear

relationship directly. P-values (and by extension confidence intervals) measure the

strength of evidence for the presence of nonzero correlation, but do not give any

indication of the strength of correlation itself. In a large sample with multiple comparisons, one could obtain very strong evidence for nonzero correlation when the linear relationship is actually quite weak, while in a small sample a high correlation might not reach significance.

2. There is no description of the Google Trends data selection criteria and collection procedure. This is an important drawback of this manuscript. All methodology steps should be reported in detail (e.g., period, region, category, web search, use of quotes for keywords with more than one word, individual searches, comparisons, etc.).

We have added to the methods section to make this clear. We state in the methods that we use the terms identified via Google Correlate to collect search hit data on. We have added more language to make it clear that terms identified from Google Correlate were inputted into Google Trends unaltered to collect search term hit data for each region. I also included text to make it clear that gtrendsR is an R interface for Google Trends that allows for an automated process of collecting search term data. 

3. This is an information epidemiology (infodemiology) study. I suggest that the authors study the relevant literature in order to gain insight and enhance their literature review. An introductory paragraph could be added in the Introduction Section.

We have included a paragraph, lines 93-102, to the introduction outlining infodemiology and its use in predicting disease and better informing the general public about health-related outcomes. 

4. The analysis (data collection) was conducted in September 2019, considering data up to December 2018. It is now 2021, and there are two more years’ data available. I believe it would add to the value of this manuscript if the analysis was updated.

We appreciate and understand the reviewers concern for having recent data for publication. However, the authors’ intention of this work is to show that value of including environmentally related features when nowcasting with Google Search terms, which does not require all recent data. This manuscript highlights the importance of considering environmental factors when creating prediction models for vector borne diseases. To this end, the models have also been posted on Eric Kontowicz’s Github (https://github.com/ekontowicz/Lyme-disease-Elastic-Net-regression-Nowcasting) for further use by other researchers and additional updates. Lastly, given the time it takes to have rigorous peer review, particularly during this COVID-19 pandemic these findings will lag all surveillance data available.

---

## [Editor Report · Decision Letter 2]

2 Feb 2022

Inclusion of environmentally themed search terms improves Elastic Net regression nowcasts of regional Lyme disease rates

PONE-D-21-11338R2

Dear Christy:

I am pleased to inform you that your manuscript has been judged scientifically suitable for publication and will be formally accepted for publication once it meets all outstanding technical requirements.

Kind regards,

Sam R. Telford III

Academic Editor

PLOS ONE

Additional Editor Comments (optional):

Christy, sorry this has taken so long.  Paradoxically with COVID and people being at home, it is no easier finding qualified reviewers and getting reviews back than when they were at work.
---

## [Editor Report · Acceptance letter]

1 Mar 2022

PONE-D-21-11338R2 

Inclusion of environmentally themed search terms improves Elastic Net regression nowcasts of regional Lyme disease rates 

Dear Dr. Petersen:

I'm pleased to inform you that your manuscript has been deemed suitable for publication in PLOS ONE. Congratulations! Your manuscript is now with our production department. 

Kind regards, 

on behalf of

Dr. Sam R. Telford III 

Academic Editor

PLOS ONE